# Poly(9*H*-carbazole) as a Organic Semiconductor for Enzymatic and Non-Enzymatic Glucose Sensors

**DOI:** 10.3390/bios10090104

**Published:** 2020-08-23

**Authors:** Gintautas Bagdžiūnas, Delianas Palinauskas

**Affiliations:** 1Institute of Biochemistry, Life Sciences Centre, Vilnius University, Sauletekio av. 7, LT-10257 Vilnius, Lithuania; delianas.palinauskas@chgf.vu.lt; 2Department of Functional Materials and Electronics, Center for Physical Sciences and Technology, Sauletekio av. 3, LT-10257 Vilnius, Lithuania

**Keywords:** Polycarbazole, glucose sensor, glucose oxidase, organic semiconductor, non-enzymatic sensors

## Abstract

Organic semiconductors and conducting polymers are the most promising next-generation conducting materials for electrochemical biosensors as the greener and cheaper alternative for electrodes based on transition metals or their oxides. Therefore, polycarbazole as the organic semiconducting polymer was electrochemically synthesized and deposited on working electrode. Structure and semiconducting properties of polycarbazole have theoretically and experimentally been analyzed and proved. For these electrochemical systems, a maximal sensitivity of 14 μA·cm^−2^·mM^−1^, a wide linear range of detection up to 5 mM, and a minimal limit of detection of around 0.2 mM were achieved. Moreover, Michaelis’s constant of these sensors depends not only on the enzyme but on the material of electrode and applied potential. The electrocatalytic mechanism and performance of the non- and enzymatic sensors based on this material as a conducting layer have been discussed by estimating pseudocapacitive and faradaic currents and by adding glucose as an analyte at the different applied potentials. In this work, the attention was focused on the electrochemical origin and mechanism involved in the non- and enzymatic oxidation and reduction of glucose.

## 1. Introduction

Excess glucose as a natural source of energy of living organisms in the blood can signal metabolic diseases such as diabetes mellitus and other cardiovascular, urinary, and nervous disorders. To prevent these diseases, a sensitive detection system is essential for clinical and biomedical applications [1]. Among detection systems such as chromatography, electrophoresis, and other spectroscopy methods, amperometric biosensors are highly desired for the medical diagnosis [2]. In turn, the amperometric biosensors are divided into three generations according to the analytical signal registration principle [3]. The first-generation glucose biosensors are based on electron transfer from cofactor of the enzyme, which is immobilized on a working electrode, to molecular oxygen as an electron acceptor in the case of oxidases and then the concentration of formed hydrogen peroxide as a reduced product is detected electrochemically [4]. In the second generation of the biosensors, artificial redox mediators are employed. These mediators can freely diffuse in a solution. In such a way, electron transfer from immobilized enzyme to working electrode may occur [5]. However, an ultimate goal of the biosensors is to eliminate the usage of the mediator toward lower fabrication cost and complexity while increasing the durability of the glucose sensor. Therefore, the third-generation biosensors based on the direct electron transfer from immobilized enzyme to the working electrode are a more progressive type of sensor. In some particular cases, the third-generation biosensors can be modified by an additional conducting material that is able to participate in the electron transfer [6,7]. Most authors claim that the native glucose oxidase does not undergo direct electron transfer because flavin adenine dinucleotide (FAD) as a cofactor binds in a deep pocket of this enzyme [6,8,9]. However, we have applied advanced theoretical and numerical approaches to predict organic semiconductor adaptability for the direct electron-transfer-based biosensors based on immobilised glucose oxidase (GOx) [10]. Based on this theoretic model, the hole-hopping from electrode to reduced flavin of GOx through tryptophan/tyrosine chains is possible when the ionization potential of the organic semiconductor is lower than 5.2 eV vs. vacuum. Moreover, electrons from the cofactor can be transferred only through a long-range direct tunnelling mechanism in a narrow potential range, which is strongly depended on the distance between the enzyme and the electrode surface [10]. On the other hand, organic semiconductors can reconcile a contact between the biological media and electronics, and these compounds-based devices have frequently been employed for sensor applications [11]. Therefore, a stable bio-sensing system based on a polymer with carbazole moieties as a hole-transporting (p-type) semiconductor and covalently attached GOx on this organic semiconductor has recently been developed [12]. Other semiconducting polymers with low ionization potential based on the electropolymerized *N*-substituted carbazole derivatives have also been applied for the GOx-based biosensors in our group [13]. Moreover, more stable electrochemical glucose sensors based on the results of our theoretical and experimental approach, which exhibited excellent electrochemical characteristics and sensing performance, have recently been developed and investigated [14,15].

However, these enzyme-based biosensors are usually expensive and not stable for a long time, and their sensing activity is affected by several factors such as temperature, pH, and ionic strength [16]. Therefore, sensors based on the non-enzymatic detection of glucose have been developed due to their low cost, rapid response, and exceptional sensitivity, as well as their direct electro-oxidation capability [17]. The development of nanotechnology has also offered new opportunities to construct nanostructured electrodes for these sensors based on different metal oxide such as ZnO, CuO/Cu_2_O, NiO, Co_3_O_4_, MnO_2_, etc., for glucose-sensing applications [18]. Recently, many other systems based on metal-organic frameworks (MOFs) and metal-azolate frameworks (MAFs), which can work under both alkaline and physiological pH conditions, have been applied for the non-enzymatic sensors [19]. Therefore, the non-enzymatic sensors, which could operate in a physiological media and without precious metals, are an important practical achievement. Moreover, organic conducting polymers or semiconductors are among the material facing the highest demand in material science due to their environmental stability, low cost, tuneable electronic properties, and easy synthesis. However, the non-enzymatic glucose sensors based on these organic materials are not fully explored so far [20]. Some examples have been reviewed here. Deshmukh et al. have deposited polyaniline and reduced graphene oxide nanocomposit functionalized with silver nanoparticles on working electrode. This nanocomposite was used for non-enzymatic sensing of glucose [21]. A potentiometric non-enzymatic glucose sensor using a molecularly imprinted layer bonded at a conducting polymer has been developed by Kim et al. [22]. Moreover, the non- and enzymatic glucose sensors based on a hierarchical Au–Ni alloy with a conductive polymer have been prepared and tested by Lee et al. [23]. Enzyme-free glucose sensors based on electrochemically prepared polypyrrole-chitosan-TiO_2_ nanocomposite films have been developed by Al-Mokaram et al. [24]. In almost all these examples of sensors, the metal oxides have been used for a nanostructurization of the electrode surface.

In the present study, organic mixed ionic/electronic conductors based on polycarbazole were synthesized and deposited on a graphite electrode. This polymer was applied for the enzymatic and non-enzymatic glucose sensing under the physiological condition. The surfaces of each sensor layer were examined using electrochemical and surface characterization techniques. To determine the optimal applied potential and sweep rate for these sensors, pseudocapacitive and faradaic currents were theoretically separated and analyzed. Based on these experimental results, the sensing mechanisms of the direct hole transfer from the enzyme to the electrode and direct glucose oxidation on the electrode were proposed. Moreover, the catalytic oxidation of the H_2_O_2_-based first-generation glucose sensor formed on the polycarbazole doped with potassium ferricyanide was compared with our other sensors.

## 2. Materials and Methods

### 2.1. Chemicals

All the analytical grade reagents such as *D*-(+)-glucose, glucose oxidase (GOx) from *Aspergillus niger* with a catalytic-activity of 208 U/mg, and a glutaraldehyde solution (25 % in water) were purchased from Sigma-Aldrich (St. Louis, MO, USA). Tetrabutylammonium hexafluorophosphate (TBAPF_6_) as a supporting electrolyte was purchased from Alfa Aesar (Kandel, Germany). Dichloromethane (DCM) as an organic solvent was distilled over CaH_2_ and stored under Ar atmosphere over 3Å molecular sieves. The solution of 0.10 M glucose was prepared in deionized water. This solution was left at least for 24 h before use to allow the mutarotation of *glucose* (i.e., to reach equilibrium between the α- and β-forms). The phosphate-buffered saline solution (PBS) of 0.1 M, pH 7.4, was used in these experiments. All the solution and buffers were prepared by using water from an ultra-pure Milli-Q Water System (Merck KGaA, Darmstadt, Germany).

### 2.2. Instrumentation

The electrochemical modification of graphitic electrodes and the measurements by cyclic voltammetry (CV) were performed by using an Autolab PGSTAT 30 (Metrohm, Utrecht, The Netherlands) potentiostat/galvanostat. which was operated by the Nova 1.10 software (Metrohm). The Ag/AgCl in 3 M KCl electrode and silver wire coated with AgCl were used as the reference electrodes in aqueous solutions and in nonaqueous media, respectively. Graphite rod (3 mm in diameter and length of 3 cm) as a working electrode was polished by abrasives and sonicated in acetone for 15 min. Sides of this electrode were covered with rubber, and only the bottom of the electrode with a surface area of 0.071 cm^2^ was used. A titanium plate with a surface area of 1 cm^2^ was used as an auxiliary electrode for the electrochemical measurements. The surfaces of the graphite rod and the polycarbazole layer were imaged by a Hitachi TM3000 (Hitachi-HighTech, Tokyo, Japan), scanning electron microscope (SEM) at 10 kV accelerating voltage in vacuum. Raman spectra of the materials were measured by a WiTec Alpha 300 R Raman (WITec, Ulm, Germany) spectrophotometer equipped with a laser of 534 nm.

### 2.3. Electrode Preparation and Investigation

The thin layer of polycarbazole (polyCz) was deposited onto the graphite surface using a cyclovoltammetric technique. The electropolymerization was performed in a solution of 1 mg·mL^−1^ 9*H*-carbazole as a monomer in dry dichloromethane using 0.1 M tetrabutylammonium hexafluorophosphate (TBAPF_6_) as the supporting electrolyte. The electropolymerization was carried out in a potential range from −0.2 V to 1.5 V versus the Ag/AgCl wire electrode at 50 mV·s^−1^ scan rate for five cycles. The energy levels in electronvolts of the deposited polyCz were calculated using the following empirical Equations (1) and (2) [25]:HOMO = *e* (*E_ox_^onset^* + 5.1)(1)
LUMO = *e* (*E_red_^onset^* + 5.1)(2)
where *E_ox_^onset^* and *E_red_^onset^* are onsets of the oxidation and reductions bands vs. Ag/AgCl, respectively, and *e* is charge of electron. The active area of polycarbazole coated graphite electrode surface was assessed by running CV experiments at sweep rates from 10 to 100 mV·s^−1^ in 5.0 mM K_3_[Fe(CN)_6_] as a redox agent and 0.1 M KCl aqueous solution. The active area of these working electrodes was calculated by using the Randles–Sevcik’s Equation (3) [26]:*I_p_* = 0.446 *nFAc*(*nFvD*/*RT*)^0.5(3)
where *I_p_* is current, *n* is the number of electrons transferred in the redox process (here, *n* = 1), *F* is a Faraday constant (96 485 s·A·mol^−1^), c is the Fe(CN)_6_^3−/4−^ concentration of 5.0 mM, *R* is an ideal gas constant (8.314 J·mol^−1^·K^−1^), *T* is an absolute temperature (298 K), *ν* is the scan rate of potential, and *A* is an active area of the working electrode. The diffusion coefficient (*D*) of 7.6 × 10^−6^ cm^2^·s^−1^ of the Fe(CN)_6_^3−^ ions was used. Current densities (*j*) were calculated using the corresponding active area of the electrode’s surfaces.

Pseudocapacitive (electrical double layer or outer surface) and faradaic (diffusion-limited or electron transfer) processes were investigated using the CV experiments of these electrodes at different sweep rates. Total currents densities (*j(ν)*) at the fixed potentials of −0.50, 0.35, and 0.80 V vs. Ag/AgCl was evaluated by using the Equation (4) [27]:*j*(*ν*) = *k_cap_* ν + *k_far_**ν*^0.5(4)
where *k_cap_* and *k_far_* are proportionality constants related to the capacitive and faradaic processes, respectively. These constants and the capacitive and faradaic current densities were estimated by nonlinear least-squares fitting of data from the CV curves at these potentials. For the data fitting, the regression Equation (4) was inputted and plotted in a Microsoft Excel (Microsoft, Redmond, WA, USA) worksheet, and the sum of squared residuals was computed and minimized by using a solver add-in to obtain the set of parameter values that best described to the experimental data [28].

The surfaces of the conducting polymer films were modified by adding 8 μL of 20 mg·mL^−1^ GOx solution in PBS (pH 7.4) onto the surface with the deposited polyCz layer and drying under ambient conditions. The GOx macromolecules that were adsorbed on the surface were crosslinked with each other by the glutaraldehyde vapour resulting from its 2% *v/v* aqueous solution and unbound GOx was washed by distilled water. The prepared electrodes were kept in a closed vessel over PBS (pH 7.4) at +4 °C until they were used in the experiments. The non-enzymatic (polyCz) and ezymatic (polyCz/GOx) sensors were tested by adding small amounts of 0.10 M aqueous glucose solution with pipetman. This glucose solution was added to the phosphate-buffered electrolyte solution to achieve the concentrations in the range from 0.1 mM to 10 mM. The reference electrode Ag/AgCl in 3 M KCl, these non-ezymatic or enzymatic electrode as the working electrode, and the Ti plate as the auxiliary electrode were used. To see the full picture of theoxidation and reduction of polyCz or the analyte in aqueous solution, two cycles were performed from −1 to 1 V vs. Ag/AgCl at a scan rate of 30 mV·s^−1^ in 0.1 M PBS (pH 7.4) at room temperature, and the second cycle of CV was used for the evaluations.

The values of Michaelis’s constants (*K_M_*) were calculated by employing the Lineweaver–Burk’s approach (Equation (5)) [13]:1/*j* = 1/*j_max_* + *K_M_*/(*j*_*max*^*c*_)(5)
where *j_max_* is the maximal current density and *c* is the concentration of glucose.

By taking signal to noise ratio of 3.0, the limits of detection (LOD) were calculated using Equation (6), where *a* is the slope and *σ* is the standard deviation of the blank. These value of *a* was determined from the linearity range of the calibration curves. The value of *σ* was calculated by measuring the currents without the analyte for 5 times at the corresponding potentials.
LOD = (3.0 *σ*)/*a*(6)

### 2.4. Details of Theoretical Calculations

The oxidized and reduced polyCz structures were optimized by using a hybrid B3LYP functional and a 6-31G(d,p) basis set with the polarization functions on H, C, and N atoms. Computational calculations of their harmonic vibrational frequencies to verify the stability of these structures was used. Vibration frequencies of the Raman spectra were estimated by employing the B3LYP functional and a 6-311G(d,p) basis set. Moreover, wavenumbers of these computed spectra were scaled by 0.966. Values of the HOMO/LUMO energy were additionally corrected using B3LYP/6-311G(d,p) level of theory with a polarizable continuum solvation model for water (*ε* = 78) for the estimation of the solvation behavior. For the Raman spectra calculations, the correction parameters of laser wavelength of 534 nm and sample temperature at 298 K were used. The values of internal reorganization energy for the modelled compounds were estimated using the four points of energy from the adiabatic potential energy surface at the B3LYP/6-31G(d,p) level of theory according to the following Equation (7) [29,30]:*λ* = [*E_±_*(*g*_0_) − *E_±_* (*g*_±_)] + [*E*_0_(*g*_±_) − *E*_0_ (*g*_0_)](7)

In this Equation, *E*_0_ and *E*_±_ are the energies of neutral (*g_o_*) and charged (*g*_±_) molecules in the geometry of cationic/anionic species (*g*_±_), respectively. All the computations were carried out by using Spartan’18 software (Spartan’18 for windows version 1.4.0 Wavefunction, Irvine, CA, USA).

## 3. Results and Discussion

### 3.1. Synthesis and Surface Characteristics of the Polycarbazole Layer

Electropolymerization is the most convenient and reliable method for the deposition of conducting polymer thin films on electrode [31]. The electropolymerization of 9*H*-carbazole on working graphite electrode was performed by using cyclic voltammetry (CV) in dry dichloromethane solution with tetrabutylammonium hexafluorophosphate as the supporting electrolyte (Appendix A). From the first scan, the oxidation of 9*H*-carbazole to its radical cation at onset of 1.15 V vs. Ag/AgCl potential was observed. According to our previous work, *N*-substituted carbazole-based monomers easily oxidized and formed 3,3’-bicarbazole via the 3,6-positions in the diffusion layer. From these dimers, oligomers and polymers can further be formed via an electrochemical oxidation reaction [13]. After the first scan, the new peak at 0.99 V vs. Ag/AgCl (at onset of 0.85 V) emerges, which can be assigned to redox process of the formed polycarbazole (polyCz). During CV scanning, the currents at 0.99 V increased until the fifth cycle and progressive deposition of the green-colored polymer on the electrode surface was indicated. Thickness of the polycarbazole layer on the graphite electrode cannot be measured with our available methods and tools. However, based on our previous work, the thickness for the other carbazole-based polymer was achieved to be around 1 μm [32].

For verification of this prepared polymer structure, the Raman spectra of polyCz, starting 9*H*-carbazole, and TBAPF_6_ were analyzed (Figure 1a). Moreover, the theoretical Raman spectra of the proposed oxidized and reduced forms of polyCz (see Figure 1a) were computed by using the hybrid B3LYP functional and the 6-311G(d,p) basis set. We proposed that this polymer exists in the oxidized (polyCz_ox) and reduced (polyCz_red) forms (Figure 1b). In the structure of polyCz_red, the 9*H*-carbazole units are connected through 3,6-C-C bonds. However, in the structure of polyCz_ox, the 9*H*-carbazole units of the polymer are oxidized toward an emeraldine-like form. This similar emeraldine-like form is typical for polyaniline (PANI) [32]. Moreover, this oxidized polyCz form was proposed by Mengoli et al. [33]. According to these experimental and theoretical Raman spectra, the traces of the electrolyte and the monomer in the synthesized polymer were not indicated because the corresponding bands of the starting compounds do not exist on the spectra of polyCz. The experimental and theoretical Raman spectra show that the structure of polyCz corresponds well to the oxidized polyCz_ox form. First of all, on the spectrum of polyCz, a band at 3065 cm^−1^, which corresponds to a stretching of N-H bond, is not indicated. This band is seen only on the Raman spectrum of the starting 9*H*-carbazole. Second, the theoretical spectrum of polyCz_ox best fits to the experimental Raman spectrum (Figure 1a). The bands at 1617, 1504, and 1236 cm^−1^ can be attributed to stretching vibrations of the C=N, C=C, and C–N bonds in the conjugated part of the oxidized carbazole moiety by the theoretical computations, respectively. Therefore, structure of the polyCz polymer can be attributed to the emeraldine-like form of polyCz_ox. Moreover, this structure of the polymer is different from the polymers based on the *N*-substituted carbazole derivatives because the stable emeraldine-like form of these polymers is not possible [13].

On the one hand, to gain more knowledge on semiconducting properties of polyCz, a theoretical and experimental study of this polymer was carried out, with emphasis on its electronic structure and possible application as hole and electron transporting (ambipolar) organic semiconductor for biosensors. We proposed that suitable values of energy levels of the highest occupied molecular orbital (HOMO) as a valence band and the lowest unoccupied molecular orbital (LUMO) as a conduction band are required to facilitate an admission of holes and electrons from the adjacent hole-transporting and electron-transporting materials, respectively, to reacting center of the immobilized enzyme or analyte in the diffusion layer of the electrode. We have previously showed that this alignment of the levels is essential for the organic electronics such as organic light-emitting diodes and organic thin-film transistors [34,35,36]. It is known that the theoretical HOMO/LUMO and corresponding experimental ionization potential (IP)/electron affinity (EA) values can be approximated via Koopmans’s theorem as IP= −HOMO and EA = −LUMO [37]. The HOMO and LUMO values of polyCz were estimated from CV of the polyCz layer modified graphite electrode in 0.1 M PBS (pH 7.4) at the scan rate of 50 mV·s^−1^ (Figure 2a). The onsets of oxidation and reduction potentials were found to be 0.41 V and −0.20 V vs. Ag/AgCl, respectively. Based on these experimental results, the energies of the approximated frontier HOMO and LUMO orbitals were calculated to be −5.5 eV and −4.9 eV, respectively. Moreover, the theoretical frontier orbitals of the polyCz_red and polyCz_ox molecules, which have four carbazole monomer units, were computed by using the DFT method. These values of HOMO/LUMO were computed to be −5.59/−4.16 eV (a band gap is 1.43 eV) and −5.23/−1.16 eV (4.07 eV) for polyCz_ox and polyCz_red, respectively. Figure 2b shows the experimental and theoretical energy levels (i.e., HOMO and LUMO). The lower band gap of the polyCz_ox molecule is narrower than for polyCz_red because the two emeraldine-like carbazole fragments in the theoretical optimized structures of polyCz_ox are connected through the double bond that increases the π-system conjugation. These theoretical results show that the polyCz_red form (i.e., the oxidation level corresponds to exp. HOMO of −5.5 eV) can electrochemically be oxidized to the polyCz_ox form. The polyCz_ox form can be reduced to polyCz_red (i.e., the reduction level corresponds to LUMO of −4.9 eV). Due to the partial doping of the oxidized polyCz_ox form with the supporting electrolyte and protonation of the N atoms of the carbazole fragments, these discrepancies between the energies of the theoretical and experimental levels can be explained.

On the other hand, the internal reorganization energy (λ) is an important factor for the determination of the semiconducting properties of organic compounds. It describes the difference of relaxation energies of a molecule from the geometry of the neutral state to the charged (i.e., radical anion and cation) state and from the charged to neutral state. For semiconductors with the highest charge carrier mobilities, the low reorganization energy is typical [38]. To compute these reorganization energies, the B3LYP/6-31G(d,p) level of DFT was used. For the polyCz_ox and poly_Cz_red molecules, the values of λ were estimated to be 0.39/0.48 eV and 0.12/0.13 eV for holes/electrons, respectively. Due to the fact that the polyCz_red is easily oxidized and polyCz_ox is easily reduced, the reorganization energies of 0.12 eV for holes and of 0.48 eV for electrons can be attributed for polyCz. From the results presented above, the ambipolar charge carrier properties for polyCz have been predicted. Moreover, Abthagir and Saraswathi found that polycarbazole exhibit electron hole conduction is similar to a conventional semiconductor [39]. It is important to note that only the hole mobility has been observed for *N*-substituted polycarbazoles by Yasutani e al. due to their low ionization potential [40]. Therefore, this polyCz polymer can be applied as the hole and electron transporting organic semiconductor for biosensors.

Figure 3 shows morphology and microstructure of the bare graphite electrode and the electrodeposited polyCz surfaces. First of all, the bare graphite electrode is sufficiently smooth, and this electrode has natural cavities with diameters of around 50 μm (Figure 3a) that are difficult to align manually with abrasives. However, after the electrochemical deposition of polyCz on the electrode, the surface has a granular structure (Figure 3b). The formation of hollow microspheres after interfacial polymerization of 9*H*-carbazole by using ammonium peroxodisulfate as an oxidizing agent was observed by Gupta and Prakash [41]. However, these structures after the electrochemical polymerization of 9*H*-carbazole were not observed. Judging from the morphology of the polyCz surface, the active surface area of the polyCz electrode must be larger than that of the bare graphite electrode. To investigate the active surface area of these electrodes, Fe(CN)_6_^3^^−^^/4^^−^ as a redox probe was used, and cyclic voltammetry (CV) was performed at different scanning rates in 0.1 M KCl aqueous solution (Figure 4a). The active area of the bare graphite and polyCz electrodes were calculated to be 0.052 cm^2^ and 0.18 cm^2^ using Equation (3), respectively. These experimental results clearly show that the active area of the electrodeposited polyCz electrode is 3.5 times higher than the area of the bare graphite electrode.

### 3.2. Electrochemical Behavior of the PolyCz and PolyCz/GOx Layers

On the electrodeposited polyCz layer, glucose oxidaze (GOx) was immobilized, and the electrodes of polyCz and polyCz/GOx were electrochemically investigated. The nature of faradaic and pseudocapacitive processes on/in these electrodes by the CV experiments at different sweep rates was investigated. Figure 4a,b shows CVs at sweep rates from 10 mV·s^−1^ until 100 mV·s^−1^. After the immobilization of GOx, the total currents from CV at these fixed potentials of −0.50, 0.35 and 0.80 V·vs Ag/AgCl were assumed as the sum of the currents for the pseudocapacitive and faradaic processes [27]. The pseudocapacitive currents appear by charging or discharging electrical double layer on the electrode surface. In turn, the diffusion-limited or faradaic currents appear by diffusing redox molecules toward the surface and/or by oxidation and reduction of the electroactive molecules on/in layer of the electrode, respectively. However, due to an absence of these redox molecules in the electrolyte solution, the faradaic currents appear due to migration of charge carriers in the semiconductor layer and charge transfer between enzyme and this semiconductor on the electrode. The migration of charge carriers in the semiconductor layer has been investigated in our previous work [42]. If the enzyme is immobilized on gold or another well conductive surface, then the currents of a direct electron transfer between the enzyme and conducting surface or redox reactions between analyte and surface only limit the faradaic currents. The pseudocapacitive and faradaic currents were estimated by nonlinear least-squares fitting data from the CV curves at these potentials. Moreover, the real surface areas of the polyCz and polyCz/GOx electrodes were employed for these calculations. The values of these calculated current densities are tabulated in Appendix A.

At the applied −0.50 V and 0.80 V potentials, the faradaic current for both electrodes limits the total currents because charge carriers migrated in the semiconductor layer at the applied electric field. For this reason, the direct charge transfer between the enzyme and surface cannot be monitored. However, the pseudocapacitive currents at −0.50 V and 0.80 V for polyCz/GOx are almost 10 times higher than these pseudocapacitive currents of the enzyme-free electrode (Appendix A). This observed difference of pseudocapacitive currents show that additional charge is injected to/from the enzyme from/to the electrical double layer on the polyCz/GOx electrode surface. This effect can also occur when the double layer thickness is increased by the additional GOx layer, which contains charged residues. Moreover, at the applied potential of 0.35 V, the similar pseudocapacitive and faradaic currents were estimated for both electrodes. Therefore, the enzymatic processes can be indicated only at 0.35 V because the pseudocapacitive currents have a greater influence for the total currents at this potential. Moreover, by using this model, the biggest differences in these currents at the range of 30–60 mV·s^−1^ of sweep rates were calculated. Unfortunately, the faradaic currents between the enzyme and electrode cannot be measured during our experiment.

### 3.3. Performance Comparison of the Enzymatic and Non-Enzymatic Sensors

The investigation of the non-enzymatic (only the polyCz coated electrode) and GOx catalyzed glucose oxidation reactions on the polyCz/GOx electrode was carried out by running CV at the physiological pH level (pH 7.4) and by using glucose, concentrations of glucose from 0.1 to 10 mM. Figure 5a,b shows that the current densities at around −0.5 and 0.35, 0.80 V decrease or increase for both electrodes by adding glucose, respectively. For the non-enzymatic polyCz-based sensor, a direct glucose oxidation and reduction at corresponding positive and negative potentials on the electrode was proposed, respectively (Figure 5c) because the electrochemical reduction of glucose to sorbitol is possible [43]. Linear ranges of these responses were found to be from 0.2 mM to 2.5, 2.0, and 3.1 mM of glucose at −0.5, 0.35, and 0.80 V, respectively. Moreover, a compliance of these titration data toward the pseudo-first order reaction kinetics was tested by using the Lineweaver–Burk’s approach. Points at higher concentration than 0.2 mM of glucose well fit to this approach (Figure 5e). For the enzymatic polyCz/GOx-based sensor, the linear ranges at −0.5 V and 0.8 V are very similar to the corresponding ranges of the non-enzymatic sensor. However, at 0.35 V, wider linear range up to 4.9 mM of glucose was found. Moreover, only at 0.8 V potential, the pseudo-first order reaction kinetic was not observed (Figure 5d).

By using the Lineweaver–Burk’s approach, the apparent Michaelis’s constants (*K_M_**) were calculated for both sensors. The asterisk indicates that these results are not real Michaelis’s constant of GOx. *K_M_* values have been found to be 0.3 mM for GOx immobilized on a microelectrode with respect to oxygen [44], and 38 mM for GOx in the solution [45]. This constant is specific to an applied electrode (i.e., not only for the enzyme). On the one hand, values of *K_M_** were estimated to be around 0.7 mM at the positive potentials and 1.6 mM at −0.5 V for the non-enzymatic electrode. On the other hand, for the enzymatic electrode, these constants are around 0.6 mM at all the applied potentials.

Another descriptive criterion of the sensors is a limit of detection (LOD), which is the lowest quantity of analyte that can be distinguished from the absence of the analyte by taking signal to noise ratio of 3.0. From the linearity range of the curves of calibration, values of LOD were calculated to be from 0.1 mM to 0.2 mM (see Table 1). Moreover, based on a linearity of the Lineweaver–Burk’s test (Figure 5e,f), lower glucose concentration than about 0.2 mM cannot be detected by using our sensors. These values are equal to the concentration of oxygen in air-saturated aqueous solution of around 0.2 mM. We proposed that oxygen molecules can react with the radical cation and anion species of polyCz in the non-enzymatic sensor and with the reduced cofactor (FADH_2_) in the enzymatic sensor. Thus, these sensors are not suitable for the detection of low glucose concentrations. These obtained characteristics of our sensors have been compared with data of other enzymatic carbazole-based and non-enzymatic conducting polymers-based sensors in the scientific literature (Table 1). The sensitivities of our sensors are higher than most of these sensors. However, the linear ranges of other sensors are wider due to the greater influence of the enzyme on currents. For this reason, analysis of real sample and detection selectivity to other analytes by using these layers was not applied.

### 3.4. Mechanistic Insights into the Origin of Glucose Sensing

After analysis of the scientific literature, Bartlett and Al-Lolage have reported that GOx as a commercial enzyme has some impurities [46]. They have predicted that the redox peaks in cyclic voltammograms at around −0.4 V vs. Ag/AgCl, which are usually attributed to charge transfer from GOx to electrode sometimes arise from free FAD, and possibly from catalase and/or other impurities, which are present in the supplied commercial enzyme, that are also adsorbed at the surface of the electrode. First of all, the possibility that some free FAD cofactor can be released from GOx and it can be oxidized/reduced on the electrode cannot be excluded because the FAD cofactor binds with apo-GOx non-covalently; therefore, this binding is relatively weak and holo-GOx tends to dissociate slowly into apo-GOx and FAD [47]. However, we did not notice any change on currents after immobilization of pure FAD on the polyCz surface. Secondly, catalase is an enzyme, which can catalyze the decomposition of hydrogen peroxide to water and oxygen. Moreover, this enzyme has a heme group as an iron active center attached to the enzyme. However, any currents change after addition of H_2_O_2_ by CV experiments by using the polyCz/GOx electrode was not indicated. Additionally, we doped the polyCz layer with potassium ferricyanide (K_3_[Fe(CN)_6_]) as the mediator of the oxidation of hydrogen peroxide [48]. GOx was immobilized on this doped polyCz-Fe electrode. On this electrode, glucose can be oxidized by FAD from GOx to gluconic acid and oxygen in the solution and H_2_O_2_ can be reduced to hydrogen peroxide (Figure 6, mechanism C) via a reaction: 2[Fe(CN)_6_]^3^^−^ + H_2_O_2_ → 2[Fe(CN)_6_]^4^^−^ + 2H^+^ + O_2_. The formed [Fe(CN)_6_]^4-^ is electrochemically oxidized on the electrode. In this case, the H_2_O_2_-based first-generation enzymatic glucose sensor was prepared. Figure 5d and Appendix A show that response to glucose of the polyCz-Fe/GOx electrode was observed. However, it is very important to mention that currents decreased after adding glucose in contrast to other our systems. The linearity range of the calibration curve from 0.2–2.0 mM and the sensitivity of −11 μA cm^−2^ mM^−1^ at this linearity range were estimated for this sensor (Table 1, entry 3). Moreover, from the Lineweaver–Burk linearity test of the polyCz-Fe/GOx sensor, all points, especially at low concentration, well fit to this model. It shows that oxygen does not compete in the reaction between the enzyme and glucose. Therefore, lower glucose concentrations than 0.2 mM can be measured. Summarizing these results, the experiments shows that hydrogen peroxide is not involved in the mechanism of the polyCz/GOx-based glucose detection. However, direct glucose oxidation to gluconic acid on the polyCz and polyCz/GOx electrodes was indicated at the 0.35 V and 0.8 V vs. Ag/AgCl potentials (Figure 6, mechanism A). A sufficiently wide range of concentrations from 1 mM to 5 mM at 0.35 V for the polyCz/GOx electrode was observed due to a direct hole transfer between the reduced FADH_2_ cofactor of GOx and polyCz as the organic semiconductor (Figure 6, mechanism B). In this mechanism B, the polyCz radical cation as a hole source can oxidize FADH_2_ to FAD^−^ radical by the single electron (hole) transfer process. From these two FAD^−^ radicals, the FAD and FADH_2_ forms can eventually be formed via a disproportion reaction [10]. Therefore, the direct oxidation of glucose and the direct hole transfer processes on the polyCz/GOx electrode were observed due to the insufficiently high ionization potential of polyCz.

## 4. Conclusions

The semiconducting organic polymer based on polycarbazole were synthesized and deposited on graphite electrode. The experimental and theoretical Raman spectra show that the structure of polycarbazole well corresponds to the oxidized emeraldine-like form. The active area of the electrodeposited polycarbazole-based electrode is 3.5 times higher than of the bare graphite electrode. Based on our theoretical and data from the literature, the properties of ambipolar charge carrier for this polymer have been predicted and discussed. These proposed polymer-based electrodes exhibited good electrocatalytic activity toward enzymatic glucose sensors with sensitivity of 14 μA cm^−2^ mM^−1^, a wide linear range of detection up to 5 mM. The greatest effect of the enzyme to currents was observed at the low applied potential of 0.35 V, when the pseudocapacitive currents were approximately equal to faradaic ones. The sensing mechanisms of direct hole transfer from the enzyme to electrode and direct glucose oxidation on the electrode have been proposed. Moreover, hydrogen peroxide does not affect to the mechanism of the glucose detection in this enzymatic biosensor.

## Figures and Tables

**Figure 1 biosensors-10-00104-f001:**
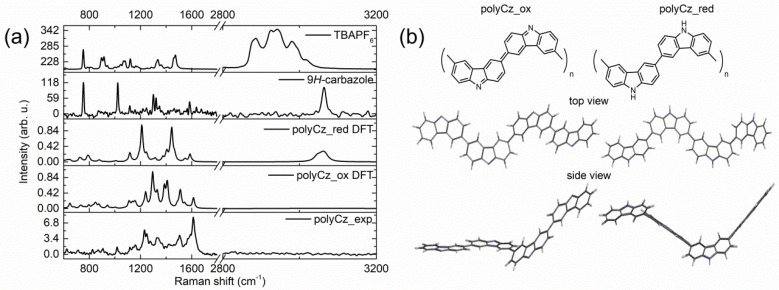
Raman spectra and proposed structures of polyCz: (**a**) Raman spectra of the prepared polyCz, theoretical spectra of the oxidized, reduced forms of polyCz, starting 9H-carbazole and TBAPF_6_ as the supporting electrolyte, respectively; (**b**) proposed structures of the oxidized (polyCz_ox) and reduced (polyCz_red) forms.

**Figure 2 biosensors-10-00104-f002:**
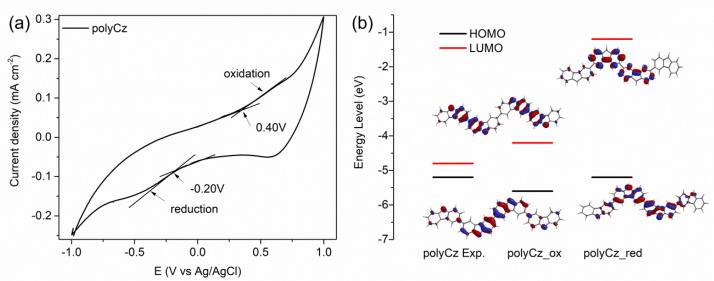
Comparison experimental and theoretical energy levels of polyCz: (**a**) cyclic voltammogram of the polyCz electrode in 0.1 M phosphate-buffered saline solution (PBS) (pH 7.4) at sweep rate of 50 mV·s^−1^; (**b**) experimental and theoretical energy levels and visualization of their orbitals (isovalue = 0.032) of polyCz.

**Figure 3 biosensors-10-00104-f003:**
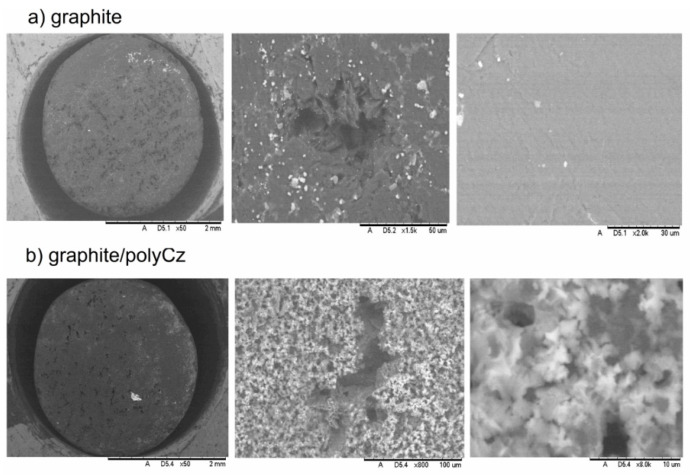
Scanning electron microscopy images of surfaces of (**a**) bare graphite; (**b**) graphite/polyCz electrodes are presented. Scales are 2 mm, 50 μm, 30 μm, 2 mm, 100 μm, and 10 μm, respectively.

**Figure 4 biosensors-10-00104-f004:**
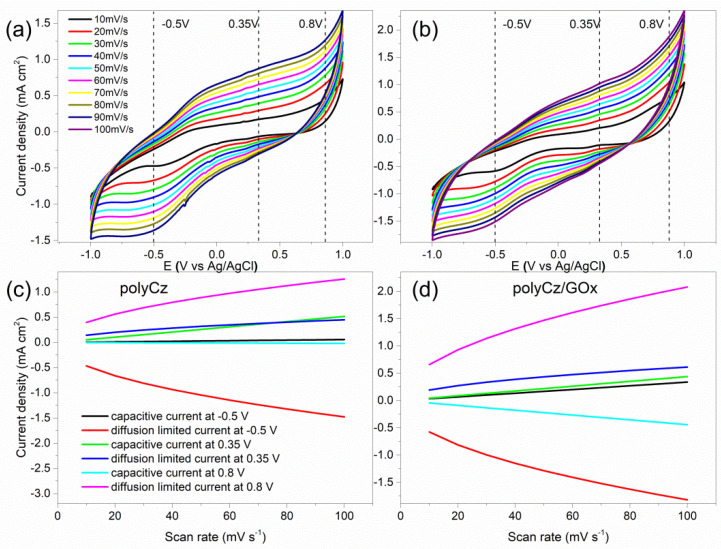
Cyclic voltammograms of (**a**) the polyCz and (**b**) polyCz/GOx electrodes in 0.1 M PBS (pH 7.4) at different sweep rates; calculated current densities of faradaic and pseudocapacitive processes at −0.5, 0.35, 0.8 V vs. Ag/AgCl on (**c**) the polyCz and (**d**) polyCz/GOx electrodes.

**Figure 5 biosensors-10-00104-f005:**
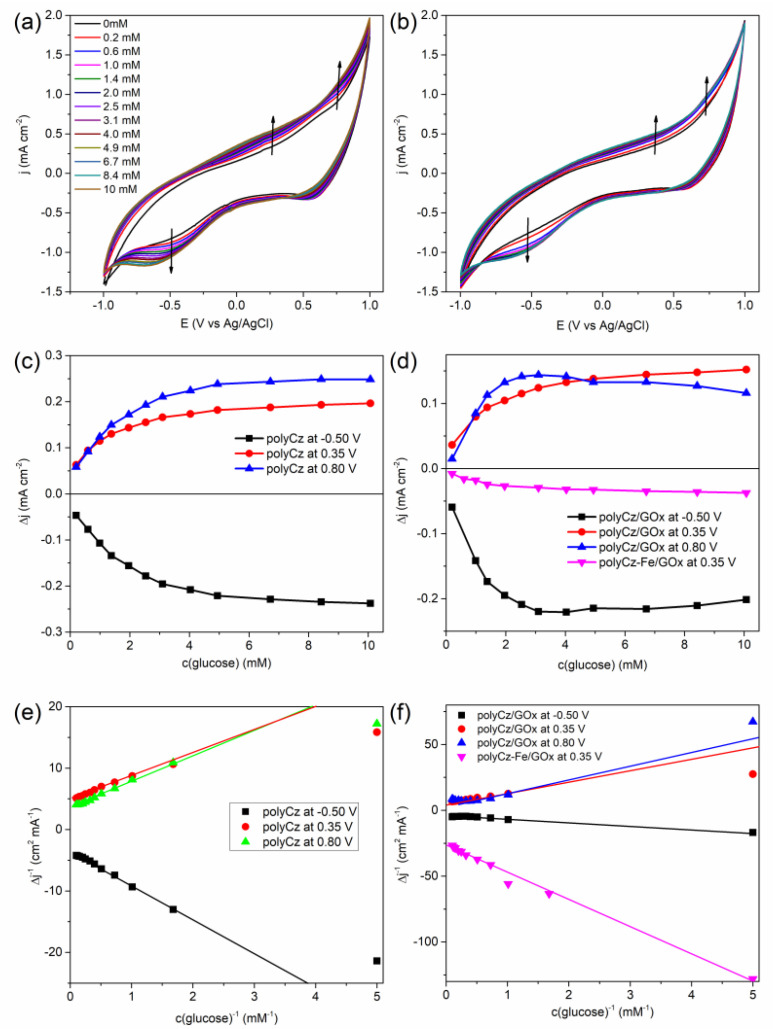
Current density responses for the non-enzymatic (polyCz) and enzymatic (polyCz/GOx and polyCz-Fe/GOx) electrodes under varying glucose concentration: (**a**) cyclic voltammetries (CVs) for the polyCz and (**b**) polyCz/GOx electrodes; (**c**) current densities response on the concentration of glucose in solution at pH 7.4 at −0.5, 0.35, 0.80 V vs. Ag/AgCl for polyCz and (**d**) polyCz(-Fe)/GOx (**d**); (**e**) the Lineweaver–Burk linearity test for polyCz and (**f**) polyCz(-Fe)/GOx.

**Figure 6 biosensors-10-00104-f006:**
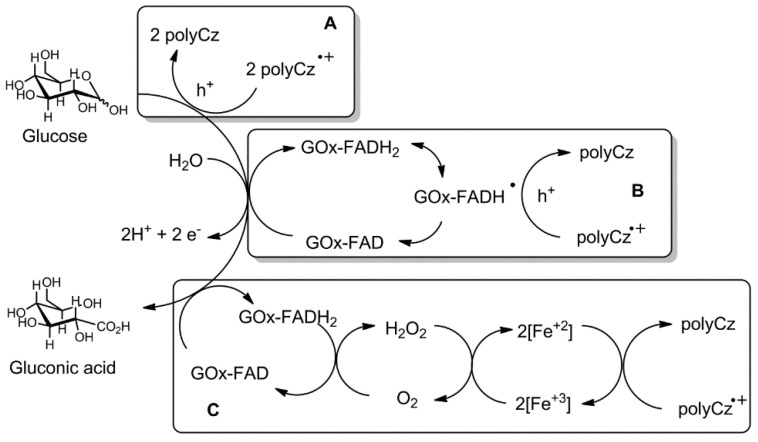
Possible glucose sensing mechanisms: **A**-direct glucose oxidation on the electrode; **B**-direct hole transfer between GOx-FAD and polyCz; **C**-H_2_O_2_-based mechanism on the poly-Fe/GOx electrode.

**Table 1 biosensors-10-00104-t001:** Comparison of our electrochemical glucose biosensors with the data available in the literature.

Entry	Active Layer of Electrode	Applied Potential, V vs. Ag/AgCl	Linear Range, mM	Average Sensitivity, μA cm^−2^ mM^−1^	Limit of Detection, mM	*K_M_^*^*, mM	Refs.
1	polyCz	−0.50	0.2–2.5	−56	0.11	1.6	This work
		+0.35	0.2–2.0	45	0.20	0.73	
		+0.80	0.2–3.1	51	0.11	0.71	
2	polyCz/GOx	−0.50	0.2–2.0	−79	0.24	0.62	This work
		+0.35	1.0–4.9	14	0.14	0.62	
		+0.80	0.2–2.0	68	0.22	N/A	
3	polyCz-Fe/GOx	+0.35	0.2–2.0	−11	0.20	0.76	This work
4	polyCzS/GOx	+0.80	2–15	0.46	0.24	−	[12]
5	polyCzEt/GOx	+0.20	1–5	3.3	0.24	2.0	[13]
6	polyCzPh/GOx	+0.20	2–5	3.7	0.23	1.1	[13]
7	PEDOT/GOx	−0.65	0.5–15	8.5	−	6.5	[14]
8	TiO_2_/PDA/GOx	+0.40	1–6	8.75	0.029	−	[15]
9	Ag–PANI/rGO	+0.80	up to 0.05	2.8	0.008	−	[21]
10	TiO_2_/CS/Ppy	+0.13	1–14	−	0.61	−	[22]

*Abbreviations:* CzS: 9-(thiiran-2-ylmethyl)-9*H*-carbazole; CzEt: 9-ethyl-9*H*-carbazole; CzPh: 9-phenyl-9H-carbazole; PEDOT: poly(3,4-ethylenedioxythiophene); PDA: polydopamine; PANI: polyaniline; rGO: reduced graphene oxide; CS: chitosan; Ppy: polypyrrole.

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
