# Peer review of "Poly(9H-carbazole) as a Organic Semiconductor for Enzymatic and Non-Enzymatic Glucose Sensors"

_biosensors, 2020, doi:10.3390/bios10090104_

Round 1
Reviewer 1 Report
This manuscript reports the preparation of polycarbazole (polyCz) through electrochemical polymerization and its sensing for glucose. Its structure and the corresponding electrochemical properties were also conducted by using cyclic voltammetry (CV), Raman spectra and SEM analyzers. The results of this article present significant scientific contributions with respect to the development of semiconducting materials. I recommend publishing after some minor revision.
1. In abstract, the data limit of detection (LOD) or KM towards glucose should be added.
2. In the electrochemical polymerization part, the thickness of polycarbazole should be provided.
3. The oxidation and reduction peaks of polycarbazole should be indicated in Figures 2b, 4a, 4b and S1.
4. In Figure 1a, there is a difference of energy bandgap (Eg) between polyCz exp. and polyCz, indicating that the conjugated length of the carbazole tetramer used by the authors was short. The authors should use longer conjugated length of the polyCz.
5. In line 274, it’s difficult to observe the cavity diameter of approximately 50 μm. The resolution of SEM images in Figure 3 was poor.
6. In lines 281-285, the statement or description “To investigate the active surface’s area….., is 3.4 times…. ” was not clear.
7. In line 315, how to define 10 times higher than the enzyme-free electrode?
There are some errors in the manuscript.
1. In lines 208 and 210, it should be Figure 1a and 1b, respectively.
2. In line 404, it should be Figure S2, not Figure S3.
Round 2
Reviewer 2 Report
see file attached

Round 3
Reviewer 2 Report
I can undesrand that the authors are not native English speaking, nevertheless, I cannot say that the English language is OK.
In any case, from a scientific point of view, the work has been further ameliorated with respect Revision 1.